# 5-Aminolevulinic Acid (5-ALA)-Induced Drought Resistance in Maize Seedling Root at Physiological and Transcriptomic Levels

**DOI:** 10.3390/ijms252312963

**Published:** 2024-12-02

**Authors:** Yaqiong Shi, Zihao Jin, Jingyi Wang, Guangkuo Zhou, Fang Wang, Yunling Peng

**Affiliations:** 1College of Agronomy, Gansu Agricultural University, Lanzhou 730070, China; 18294126077@163.com (Y.S.); jinzihao9290@163.com (Z.J.); wjy126139@163.com (J.W.); m17899314410@163.com (G.Z.); pengyl@gsau.edu.cn (Y.P.); 2State Key Laboratory of Aridland Crop Science, Lanzhou 730070, China

**Keywords:** drought stress, exogenous phytohormones, root system, weighted gene co-expression network analysis

## Abstract

Drought stress seriously affects the growth, development, yield, and quality of maize. This study aimed to investigate the effects of exogenous 5-ALA on root morphology and physiological changes in maize seedlings and to detect its regulatory network. The results showed that adding 25 mg/L 5-ALA accelerated root morphogenesis (root average diameter, main root length, total root length, and root surface area) and promoted dry matter accumulation and free radical removal. Transcriptome analysis showed that after applying exogenous 5-ALA, differently expressed genes (DEGs) were mainly involved in histidine metabolism, amino acid biosynthesis, plasma membrane components, secondary active sulfate transmembrane transporter activity, and anion reverse transporter activity. Two inbred lines specifically responded to organelle and structural molecular activity, and 5-ALA may regulate maize roots to achieve drought tolerance through these two pathways. In addition, candidate genes that may regulate maize root growth were screened by weighted gene co-expression network analysis (WGCNA). These genes may play important roles in alleviating drought stress through lignin synthesis, heat shock proteins, iron storage and transport, calcium binding proteins, and plasma membrane regulation of exogenous regulator 5-ALA. Our results may provide a theoretical basis for clarifying the response of maize seedling roots to drought and the mechanism of exogenous hormones in alleviating drought.

## 1. Introduction

Maize (*Zea mays* L.) is an important feed, industrial processing raw material, and food crop. Worldwide, the planting area and yield of maize are influenced by natural conditions such as geography, climate, and soil in different countries and regions. Especially in some important maize-producing countries such as America, China, Brazil, and Argentina. Maize is widely planted and plays a crucial role in agricultural yield, contributing significantly to global food production. The maize production in these regions is not only related to domestic food security and agricultural economy but also has a significant impact on the global food market. Maize cultivation is influenced by natural conditions such as climate, soil, and water resources. Suitable climate and soil conditions, such as a warm climate, fertile soil, and sufficient water, are key factors for maize growth. Due to insufficient or uneven rainfall, or a mismatch between the timing of irrigation and the critical water demand period of maize, there can be adverse effects on maize growth [1]. Drought is an important limiting factor that leads to unstable and low maize yields, seriously affecting the further development of maize production, and also an important factor that restricts its productivity. Enhancing the drought resistance of maize has become one of the key issues that urgently need to be addressed in modern agriculture [2]. At the biological and cellular levels, drought resistance is mainly determined by a series of complex physiological and metabolic mechanisms. This includes osmotic regulation mechanisms, which maintain intracellular osmotic pressure balance by accumulating small-molecule substances. An appropriate cation balance is required to maintain the stability of the cell membrane and ion balance inside and outside the cell. The synthesis and transport mechanisms of small-molecule substances, as well as the protection of cell membrane integrity. These mechanisms work together to enable plants to maintain normal physiological functions and growth under drought conditions. Studying the effects of drought stress on plants and their own response mechanisms to drought, utilizing genetic engineering technology to alter the physiological and biochemical characteristics of plants, and improving their drought resistance is of great significance for minimizing the damage caused by drought stress and promoting the growth and production of plants in arid areas.

The root system is an important nutrient organ that maintains the normal growth and development of plants. Its main function is to obtain nutrients and water from the soil to meet the needs of above-ground growth and development. The development of plant roots is closely related to hormone regulation, including auxin [3], cytokinin [4], ethylene [5], abscisic acid [6], and gibberellin [7]. 5-ALA is a hydrocarbon containing oxygen and nitrogen; in higher plants, 5-ALA acts as a linear five-carbon complex, and it is a key precursor in the biosynthesis of all porphyrin compounds (such as chlorophyll, ferrous heme, drill amine, photosensitive hormone, etc.). It has been shown that 5-ALA is a metabolic product within the plant, it has a certain regulatory effect on the growth of plants, and is a potential plant growth regulator [8]. Bindu and Vinekandan [9] first discovered that 5-ALA can not only induce the differentiation of adventitious roots and buds in cowpea callus tissue but also promote the growth of cowpea adventitious roots at concentrations of 7~10 mg/L. They believe that 5-ALA has dual functions as a cytokinin and an auxin. Wang et al. [10,11] observed that external application of 5-ALA can promote the transport of carbohydrates to the underground, thereby promoting the growth of cabbage and radish roots and increasing their biological yields. Liu’s [12] results indicated that 0.1–1 mg/L 5-ALA could enhance the water-deficit stress tolerance of oilseed seedlings by improving biomass accumulation, maintaining relatively high ratios of GSH/GSSG and ASA/DHA, enhancing the activities of specific antioxidant enzymes, and inducing the expression of specific antioxidant enzyme genes. Under salt stress conditions, exogenous application of 5-ALA significantly increased the root dry weight of Jinchun4 and Jinyou1 compared to NaCl stress treatment alone, and he believes that this may be related to the fact that 5-ALA can promote photosynthesis [13]. An et al. [14] treated *Arabidopsis* seedlings with 5-ALA and measured the gene expression levels of root growth, auxin synthesis, signal transduction, and transport. They demonstrated that 5-ALA significantly promoted the elongation of primary roots in *Arabidopsis* and considered that this may be due to the involvement of auxin transport in the regulation of root growth by 5-ALA. The research results of Rao et al. [15] showed that exogenous ABA treatment inhibited the elongation and growth of strawberry roots, while 5-ALA promoted the polar transport of IAA through the expression of the IAA polar transport protein *PIN1* gene to alleviate the inhibition of ABA on strawberry root growth. In addition, 5-ALA improves the water use efficiency of plants by promoting root growth, increasing the root surface area and root hair density, and interacting with other plant hormones to jointly regulate plant growth and metabolic processes and improve plant stress resistance. However, these reports have not elucidated the mechanism of action of 5-ALA on root growth and development.

As a model plant, maize has a relatively clear genome structure, which provides convenience for studying its drought resistance mechanism. By studying the physiological and transcriptomic levels of maize seedling roots, we can delve deeply into the impact mechanism of 5-ALA on their drought resistance. This provides a vital basis for exploring maize’s drought-tolerant genes and overall mechanisms, which are crucial for enhancing the crop’s resilience to drought conditions. RNA-seq (RNA sequencing) is used for comprehensive transcriptome sequencing and analysis [16], which can quantitatively analyze the transcriptome at the level of single cells, reveal gene expression differences in different cell types and states, and help accurately mine maize drought resistance genes and phenotypes. Therefore, in this study, drought-tolerant inbred line Zheng58 and drought-sensitive inbred line TS141 were used as experimental materials, and water deficiency induced by polyethylene glycol (PEG6000) was used as a stress condition. Maize seedlings were treated with exogenous 5-ALA to explore the effects on morphological and physiological changes in roots. Through transcriptome analysis, the gene regulatory network of maize seedling roots after 5-ALA pretreatment was clarified to further understand the molecular mechanism of the effect of 5-ALA on maize seedling roots and to provide a certain basis for the discovery of maize drought tolerance genes and the study of drought resistance mechanisms.

## 2. Results

### 2.1. Effects of Exogenous 5-ALA on Maize Seedling Growth

As can be seen from Table 1, drought stress inhibited the growth of maize seedlings. The seedling length (SL) of Zheng58 was significantly lower than that of CK. The seedling fresh weight (SFW), shoot dry weight (SDW), and plant biomass of TS141 were significantly lower than those of CK. Compared to drought stress, 5-ALA application significantly increased the SL and SFW of Zheng58 by 20.15% and 56.76% (*p* < 0.05). Shoot dry weight and plant biomass increased by 12.86% and 3.39%, respectively. The SFW, SDW, and plant biomass accumulation of TS141 increased significantly by 74.63%, 84.78%, and 76.54%, while SL increased by 7.59%. These results indicated that the external application of 5-ALA alleviates the inhibition of seedling growth and promotes the accumulation of biomass.

### 2.2. Effect of Exogenous 5-ALA on the Root Growth of Maize Seedlings

Plant root growth is inhibited under drought stress, but different crops or varieties adapt to drought differently. Some crops increase taproot depth or lateral root number to improve the efficiency of water use in deep soil, and some crops reduce cell elongation and increase cell division to lower the water potential gradient. As can be seen from Table 2, drought stress inhibited the root growth of maize seedlings, and the taproot length of two inbred lines decreased. The application of 5-ALA can promote the root establishment of maize seedlings and alleviate the root growth retardation caused by drought. After applying 25 mg/L exogenous 5-ALA, compared with drought stress (PEG), drought-tolerant inbred line Zheng58 increased its root dry weight by 24.43%, main root length by 45.44%, total root length by 30.7%, and root surface area by 34.7%, all of which reached significant levels, and the root average diameter also increased. The main root length, total root length, and root surface area of TS141 increased by 33.50%, 16.7%, and 17.6%, and the root dry weight and root average diameter also increased. This indicates that 5-ALA can promote the root growth of two inbred lines with differing drought tolerance (Figure 1), but the effect on the root growth of Zheng58 was greater than that of TS141.

### 2.3. Effects of Exogenous 5-ALA on the Root/Shoot Ratio, Lateral Root Number, and Relative Water Content (RWC) of Maize Seedlings

As shown in Figure 2, the root/shoot ratio of Zheng58 was higher than that of TS141, and compared that of with CK, the root/shoot ratio of TS141 had no significant change under drought stress, while the root shoot/ratio of TS141 increased by 148.01%, and the difference reached a significant level. Compared with PEG treatment, the root/shoot ratios of Zheng58 and TS141 inbred lines under AP treatment decreased by 6.57% and 44.39%, respectively (*p* < 0.05). After 5-ALA was applied, the lateral root number of inbred lines Zheng58 and TS141 increased by 33.34% and 70.59%, respectively.

As can be seen from Figure 2, the RWC of seedling roots of drought-tolerant inbred line Zheng58 showed a small overall variation, and the RWC with PEG treatment was lower than that with CK treatment, with a decrease of 10.84% (*p* > 0.05). The RWC of TS141 decreased by 15.13% (*p* < 0.05) compared with that of CK after PEG stress. These results indicated that drought stress caused relative root wilting, but there were differences among different resistant inbred lines, and the decrease for Zheng58 was less than that for TS141. Under AP treatment, the RWCs of two inbred lines were close to that of CK, but the difference was not significant. The results indicated that drought stress broke the water balance of maize seedlings’ roots, and the application of 5-ALA could help the plants maintain the balance and ensure normal root function.

### 2.4. Effects of Exogenous 5-ALA on the Root Physiological Characteristics of Maize Seedlings

As can be seen from Figure 3, the contents of root osmotic regulatory substances increased after drought stress. Compared with those in CK, the proline content (Pro), malondialdehyde (MDA) content, and relative electrical conductivity (REC) of roots of drought-tolerant inbred line Zheng58 were increased by 91.01%, 91.01%, and 19.72%, respectively (*p* < 0.05). The Pro content, MDA content, and REC of roots of TS141 were increased by 59.51%, 112.7%, and 20.48%, respectively (*p* < 0.05), and the increase in osmoregulatory substances of Zheng58 was significantly higher than that of TS141.This is a self-regulation phenomenon of plant resistance to adversity, indicating that drought stress promotes Pro production in the roots of maize seedlings to enhance their drought resistance. After AP treatment, the content of osmoregulatory substances in both inbred lines showed a decreasing trend. Compared with those of PEG, the Pro content, MDA content, and REC of Zheng58 root decreased by 36.47%, 36.47%, and 7.06% (*p* < 0.05), but the TS141 roots showed a decrease of 19.57%, 19.57%, and 7.45% respectively (*p* < 0.05), indicating that 5-ALA could reduce the damage of drought stress and reduce the exosmosis of root cells.

As can be seen from Figure 3, the superoxide dismutase (SOD), peroxidase enzyme (POD), and catalase (CAT) activities of maize seedlings significantly increased after drought stress. Compared with those of CK, the SOD, POD, and CAT activities of inbred line Zheng58 increased by 6.64%, 28.01%, and 206.45%, respectively. Inbred line TS141’s activities increased by 37.77%, 31.30%, and 270.27%, respectively, and all reached significant levels. After treatment by 25 mg/L 5-ALA, the SOD, POD, and CAT activities of maize seedling roots were further increased and were significantly higher than those with PEG treatment. Compared with those following PEG treatment, the SOD, POD, and CAT activities of Zheng58 were increased by 2.08%, 37.99%, and 68.42%, respectively (*p* < 0.05). The levels of TS141 increased by 16.20%, 66.54%, and 92.70% (*p* < 0.05). The results indicated that the application of 5-ALA could promote the reduction of oxygen free radicals, reduce oxidative damage, and alleviate the damage to maize seedlings caused by drought.

### 2.5. Sequencing Results and Analysis

#### 2.5.1. Statistics of Sequencing Results

Transcriptome sequencing was performed on the constructed library using the Illumina platform, and transcriptome analysis of 18 samples was completed. A total of 170.09 Gb Clean Data was obtained, with the Clean Data of all samples reaching 7.72 Gb, and the percentage of Q30 base was 90.64% or above with filtering out of sequences that encode peptides that are too short (fewer than 50 amino acid residues) or contain only a single exon. A total of 9218 new genes were discovered, and the amount of sample sequencing data is shown in Appendix A.

#### 2.5.2. Repeated Relevance Assessment

There is biological variability in gene expression between different individuals. The variable degree of expression of different genes is different. To find true differentially expressed genes, the expression differences caused by biological variability need to be considered. Three biological replicates were established. Spearman’s Correlation Coefficient was used as an evaluation index of biological repeat correlations. As shown in Figure 4, the replications ρ are close in pairs, indicating a strong correlation between sample replications.

#### 2.5.3. Differential Gene Expression Gene Statistics and Analysis

To comprehensively study drought tolerance of maize and the effects of exogenous 5-ALA on drought stress alleviation of maize, FPKM (fragments per kilobase per million) was used to calculate gene expression levels, and genes with differential multiples of gene expression |log_2_FC| ≥ 2 and *p* ≤ 0.05 were defined as DEGs. As can be seen from Figure 5, under CK and PEG stress, 2070 differentially expressed genes of maize inbred line TS141 were screened, among which 1053 differentially expressed genes were upregulated, accounting for 50.87% of the expressed genes. There were 1128 expressed genes in Zheng58, among which 405 differential genes were upregulated, accounting for 35.90% of the expressed genes. Compared to the value with PEG treatment, there were 3036 differentially expressed genes of TS141 after 5ALA treatment, among which 1765 genes were upregulated, accounting for 58.14% of the total expressed genes. There were 3853 differentially expressed genes in Zheng58, among which 1428 genes were up-regulated, accounting for 37.06% of the expressed genes. After the application of 5-ALA, the gene expression of the two inbred lines increased, and Zheng58 enhanced its drought tolerance by having more downregulating DEGs than TS141.

The distribution of expressed and specific differential genes in the Venn diagram also fully demonstrated the effect of exogenous 5-ALA on the root growth of maize inbred lines with different drought tolerance. As shown in Figure 5a, with CK as the control, 42 genes were expressed in the two maize inbred lines under drought conditions. With PEG as the control, 396 genes were expressed in the two inbred lines after 5-ALA was applied. With CK as the control, 41 upregulated genes were expressed in the two maize inbred lines under PEG treatment. With PEG as the control, 100 upregulated genes were expressed in the two inbred lines after the application of 5-ALA (Figure 5b). With CK as the control, 45 downregulated genes were expressed in two maize inbred lines under PEG treatment. In contrast to the drought treatment (PEG), 171 downregulated genes were expressed in the two inbred lines after the application of 5-ALA (Figure 5c).

#### 2.5.4. Gene Ontology (GO) Enrichment Analysis of Differentially Expressed Genes (DGEs)

##### Go Analysis of Two Inbred Lines

GO functional annotation is a gene function analysis method that aims to classify and annotate gene sequences to better understand their functions and the biological processes involved and covers three aspects: biological processes, cell components, and molecular functions. As shown in Figure 6, GO function analysis was performed on DEGs in two maize inbred lines. After PEG treatment, the biological process (BP) of DEG co-enrichment was mainly a cellular process, followed by a metabolic process and a single-organism process. The main enriched cell components (CCs) were cells, cell parts, organelles, and membranes. The main enriched molecular functions (MFs) were binding and catalytic activity. In the inbred line TS141, DEGs were also enriched on the transporter activity of MF. DEGs of the inbred line Zheng58 were also enriched on the membrane part of CC. The genes related to the material metabolism, cell activity, signaling pathways, and enzyme activity of maize inbred lines under drought were identified.

##### Go Analysis of Two Inbred Lines Treated with Exogenous 5-ALA

The DEGs identified by two inbred lines under AP treatment were analyzed with GO function annotation (Figure 6). The main BP of DEG enrichment in the two inbred lines under AP treatment was a cellular process, followed by a metabolic process and a single-component process. CC was mainly enriched in cells, cell parts, organelles, membranes, organelle parts, and membrane–membrane parts. The main enrichment of MF was binding and catalytic activity. The DEGs of TS141 were also enriched in the structural molecule activity of MF. In the inbred line Zheng58, the concentration of CC in the membrane was greater than that in the organelle. After the application of 5-ALA, exogenous 5-ALA enhanced the drought tolerance of maize mainly by upregulating or downregulating more DEGs.

#### 2.5.5. Kyoto Encyclopedia of Genes and Genomes (KEGG) Enrichment Analysis of DEGs

The KEGG database integrates information about genes, chemicals, and biological systems to predict the metabolic pathways and functions of genes in cells and to identify the complex biological processes in which genes are involved. The cellular processes, environmental information processing, genetic information processing, metabolism, and organismal systems involved in the two inbred lines were further investigated by KEGG metabolic pathway analysis, with five branches processed. As can be seen from Figure 7, compared with those of CK, a total of 1081 genes in TS141 were involved in 121 KEGG metabolic pathways. The most involved unigenes were plant–pathogen interaction, plant hormone signal transduction, phenylpropanoid biosynthesis, ribosome, MAPK signaling pathway–plant, starch, and sucrose metabolism (Figure 7a). Compared with those of CK, a total of 456 genes were involved in 103 KEGG metabolic pathways in Zheng58. The most implicated genes were hormone signal transduction, plant–pathogen interaction, phenylpropionic acid biosynthesis, plant MAPK signaling pathway, and starch and sucrose metabolism. Also, screening in the two inbred lines revealed carbon metabolism, amino sugar and nucleotide sugar metabolism, endocytosis and peroxidase, and other metabolic pathways (Figure 7b).

Compared with observations following PEG treatment, there were a total of 1863 genes involved in 126 KEGG metabolic pathways in TS141 after the application of 5-ALA. The most implicated genes were ribosome, hormone signal transduction, phenylpropionic acid biosynthesis, carbon metabolism, plant–pathogen interaction, and starch and sucrose metabolism. A total of 2022 genes in the inbred line Zheng58 were involved in 127 KEGG metabolic pathways. The most involved unigenes were hormone signal transduction, plant–pathogen interaction, plant MAPK signaling pathway, phenylpropionic acid biosynthesis, starch and sucrose metabolism, and biosynthesis of amino acids. The two inbred lines were involved in phosphonate and phosphinate metabolism, indole alkaloid biosynthesis, and basal transcription factor. There were significant differences in expression on 11 KEGG pathways, including transcription factors (Figure 7c,d).

### 2.6. Validation of DEGs by qRT-PCR Analysis

Five differentially expressed genes were selected, and their expression patterns were further verified and analyzed by real-time fluorescence quantitative PCR. As shown in Figure 8, the expression levels of 5 genes in the 6 groups were consistent with the expression trends of the analyses with qRT-PCR and RNA-seq, indicating that the sequencing results of this study were reliable.

### 2.7. Weighted Gene Co-Expression Network Analysis (WGCNA)

#### 2.7.1. Construction of Gene Co-Expression Modules Based on WGCNA

WGCNA can construct co-expressed gene modules by constructing co-expression networks, explore the correlations between co-expressed gene modules and sample traits, and mine the hub genes that are highly correlated with traits. To clarify the correlation between genes obtained from RNA-seq and maize drought resistance phenotype data, we constructed a gene co-expression network using WGCNA and identified nine distinct co-expression modules (Figure 9a,b). Among them, the main root length was significantly positively correlated with the MEyellow and MEgreen modules, and the total root length was significantly positively correlated with the MEgreen modules. The root/shoot ratio was positively correlated with the MEturquoise module. The number of lateral roots was significantly correlated with the MEgreenyellow, MEgreen, and MEbrown modules (Figure 9c,d).

#### 2.7.2. Gene Functional Analysis of Related Modules

To further investigate the function of drought resistance-related genes, GO and KEGG were used to analyze genes in the yellow and MEturquoise modules, which were most associated with taproot length and the root/shoot ratio. In the yellow module, the GO-rich module gene BP included metabolic processes, cellular processes, individual organism processes, biological regulation, and response to stimuli. The enriched CC included the cell, cell part, organelle, membrane, membrane part, and organelle part. The enriched MF included binding, catalytic, antioxidant, and transport activities (Figure 10a). The metabolic pathway of KEGG is mainly phenylpropanoid biosynthesis, plant hormone signal transduction, starch and sucrose metabolism, and plant–pathogen interaction (Figure 10b). In the MEturquoise module, the GO-rich module genes’ BP included metabolic process, cellular process, single-organism process, biological regulation, and response to stimulus. Enriched CCs included cell, cell part, organelle, membrane, membrane part, and extracellular region. Enriched MFs included binding, catalytic activity, and transporter activity (Figure 10c). The KEGG metabolic pathway was mainly plant hormone signal transduction, phenylpropanoid biosynthesis, carbon metabolism, sphingolipid metabolism, RNA transport, ribosome, alanine, aspartate and glutamate metabolism, valine, leucine, and isoleucine degradation, amino sugar and nucleotide sugar metabolism, and ascorbate and aldarate metabolism (Figure 10d).

#### 2.7.3. Analysis of Hub Gene Interaction Networks in the Modules

In this study, gene network visualization and gene connectivity analysis were performed for the yellow-module genes and turquoise-module genes. In order to identify the hub genes associated with maize drought resistance, we classified the 30 genes with the highest kME value in the yellow module as hub genes and utilized the hub genes and their interaction gene mapping gene co-expression network (Figure 11). In the yellow (0.89) module, the central gene was putative cinnamyl alcohol dehydrogenase 1, which is the key enzyme in lignin synthesis. Fatty acid alpha-dioxygenas are hemoproteins in the myeloperoxidase family, 17.0 kDa class II heat shock protein-like is a class II small heat shock protein gene, which has positive roles in both biotic and abiotic resistance. Vacuolar iron transporter 1.2-like participates in the storage and transportation of iron, and plays an important role in plant photosynthesis, nitrogen fixation, respiration, DNA and hormone synthesis. Polygalacturonase inhibitors can degrade pectin in plant cell walls, destroy the integrity of cells, and provide a lot of nutrients for the growth and development of pathogenic bacteria. *Zm00001d009647* is an uncharacterized gene (Figure 11a). In the turquoise (0.86) module, as a special member of the calcium-binding protein family, the central gene calcium-binding EF-hand protein and calcium ions participate in functional regulation from cell proliferation to apoptosis, playing a variety of roles in plant responses to biotic and abiotic stresses and in the development process. Tonoplast intrinsic protein 3 can improve the permeability of the cytoplasm membrane and vacuole membrane. *Zea_mays_newGene_1678*, *Zea_mays_newGene_15254*, and *Zea_mays_newGene_5352* are new genes. *Zm00001d027742* is uncharacterized gene (Figure 11b). Therefore, it is speculated that lignin synthesis, heat shock protein, iron storage and transport, calcium ion binding protein, and the plasma membrane play an important role in the drought tolerance of maize, and that the exogenous regulator 5-ALA alleviates drought stress.

## 3. Discussion

### 3.1. Analysis of the Root Growth and Physiological Characteristics of Maize Seedlings Under Drought Stress After the Application of 5-ALA

Water deficit is an important factor leading to low and unstable maize yield [17]. It is difficult for plants to absorb enough water and nutrients under drought, which makes it difficult for plants to accumulate organic matter, affecting normal growth and development [18]. Turner [19] and Subbarao [20] found that root biomass and distribution were the main drought resistance traits affecting yield. In this study, compared with that under normal treatment, the seedling biomass of maize inbred lines TS141 and Zheng58 decreased significantly after drought stress, and the root surface area, dry root weight, total root length, and root/shoot ratio all decreased. Compared with those under drought stress, the root surface area, biomass, lateral root number, total root length, and root/shoot ratio of maize seedlings were significantly increased after adding exogenous 5-ALA, which provided a morphological basis for maize seedlings’ response to drought stress.

Plant growth is inseparable from material transport and information transfer between roots and shoots [21]. Many plants change their total dry matter and root/shoot ratio by adjusting their material distribution, effectively improving their adaptability to drought stress [22]. Lateral root development is regulated by internal signals, soil water and nutrient status, and other internal and external factors [23]. According to the study of Ehdaie et al. [24], developed roots are conducive to absorbing water and nutrients in soil under drought conditions. The maize root system consists of a primary root, an adventitious root, and a lateral root at all levels. The main function of the root system at the seedling stage is completed by the main root and lateral root, and the growth and development of lateral roots has a direct impact on the growth of seedlings. In this experiment, the root/shoot ratio of Zheng58 decreased after drought treatment, which was consistent with the results of Yamauchi [25] in that drought inhibited both root expansion and above-ground water use. After TS141 encountered drought, the root/shoot ratio increased significantly. This is consistent with Mu’s research [26], which showed that drought inhibits above-ground growth, promotes root elongation, increases root/shoot ratio, and improves water use efficiency. After the addition of exogenous 5-ALA, the lateral root numbers of the two inbred lines increased significantly. The root/shoot ratio decreased significantly, indicating that 5-ALA could increase the lateral root number of maize and alleviate the increase in the root/shoot ratio caused by drought stress. This is consistent with the research results of Li et al. [27], who showed that spraying exogenous 5-ALA on the seedlings of three tree species promoted the accumulation of materials in the ground and underground parts of the seedlings and made the root system extend to adapt to drought.

RWC is used to measure the water status of plants. Drought stress can easily lead to root damage or wilting. Plants can improve cell osmotic pressure and relative water content by regulating the synthesis and accumulation of osmotic regulatory substances to maintain their normal growth and physiological functions [28]. Li [29] showed that there was an inverse ratio between RWC and stress in maize, and excessive water loss caused difficultly in maintaining chloroplast structure and PSⅡ function, which restricted photosynthesis. The results showed that the roots of the two maize inbred lines had difficulty in water absorption, the water deficit of plant tissue was serious, and the RWC was decreased and greatly affected by stress. The RWCs of the two inbred lines increased significantly after the application of 5-ALA, which was close to normal treatment. This indicates that 5-ALA can promote root water absorption and maintain the integrity of its PSⅡ function so that plants can maintain a higher level of photosynthesis and effectively repair the damage caused by drought stress. This is consistent with the results of Lu et al. [30] showing that the root relative water contents of 24 maize inbred lines decreased, and the RWCs recovered somewhat after rehydration; the results indicated that the application of 5-ALA could rehydrate the water.

Proline is an important osmoregulatory substance in plants, and plants adapt to drought stress by regulating the synthesis and accumulation of proline [31]. Proline can reduce water loss and maintain protein structural activity and plasma membrane integrity [32]. The results of this experiment showed that the proline content of maize seedlings root increased after drought treatment, and the increase of Zheng58 was higher than that of TS141. As an important osmoregulator, proline actively responds to the physiological mechanism of root stress in maize seedlings and alleviates intracellular and extracellular osmotic stress after water loss. This is consistent with the findings of Hao et al. [33] showing that the proline content of cucumber seedlings increased significantly under drought conditions, and the proline content following treatment with a certain concentration of SA after continuous drought stress was lower than that without SA stress. MDA is one of the products of lipid peroxidation in plant cells, which can affect the structure and fluidity of cells and further aggravate damage to cell membranes [34]. When the permeability of the plant cell membrane changes, the mobility of ions in the cell changes, resulting in changes in electrical conductivity. Under drought conditions, the water in plant cells is reduced, which leads to the lipid peroxidation of the cell membrane to produce MDA [35]. Both electrical conductivity and MDA are indicators used to measure plant stress damage, and stability is associated with drought resistance [36]. The results of this experiment showed that the relative electrical conductivity of the roots of the two inbred maize seedlings’ roots was higher, the cell contents were extravasated, and the plasma membrane was seriously damaged. The root REC decreased significantly after application of 5-ALA, and the variation of Zheng58 was smaller than that of TS141. The variation amplitude of MDA content was similar to that of relative conductivity, which increased after drought stress but decreased after 5-ALA application. This was consistent with the results of Pei et al. [37] who measured the relative electrical conductivity of 67 maize seedlings and found that it was significantly negatively correlated with drought tolerance. The results showed that drought stress damaged the relative permeability of the plasma membrane of maize seedlings’ roots. In addition, the application of 5-ALA could reduce the relative electrical conductivity and MDA content of maize seedlings’ roots and alleviate plasma membrane penetration.

When the balance of reactive oxygen species removal in plants is broken, excessive accumulation of reactive oxygen species occurs, resulting in a series of oxidative stress reactions such as cell membrane peroxidation, protein denaturation, and DNA damage, which affect plant growth, development, and yield. SOD, POD, and CAT can remove superoxide anions produced by environmental stress, reduce oxidative damage and ROS accumulation in cells, protect cells from oxidative damage, and maintain the intracellular redox dynamic balance [38]. In this experiment, the CAT, POD, and SOD activities of maize seedling roots were determined. The results showed that compared with those following normal treatment of CK, the enzyme activities related to the ROS scavenging system showed an increasing trend after drought stress. These results indicated that maize seedlings could cope with a large number of reactive oxygen species produced by drought stress by maintaining high protective enzyme activity and reducing the toxic effect of oxygen free radicals on root cells. After the external application of 5-ALA, the activities of CAT, POD, and SOD in inbred lines continued to increase, indicating that exogenous 5-ALA could further improve the antioxidant system activity of maize seedlings roots, maintain the metabolic balance of ROS, and eliminate drought-induced oxidative stress. This is consistent with the results obtained by Zhao [39] showing that exogenous hormones sprayed on maize under drought and rehydration can improve photosynthetic efficiency and antioxidant capacity and promote plant growth.

The difference in drought resistance between different crops and different varieties of the same crop is also different, which is related to the genetic characteristics and morphological structures of the varieties. Under normal conditions, the above-ground growth levels of maize inbred lines Zheng58 and TS141 were similar; TS141 developed taproots and a large space for root ligation, Zheng58 developed lateral roots and large root crowns, and the metabolic intensity, production, and system activity were higher for Zheng58 than for TS141. Drought stress seriously affected the normal growth of the two inbred lines. Zheng58 can promote the smooth progress of physiological and biochemical processes and avoid drought stress by improving the activity of the antioxidant system and maintaining cell structure. TS141 mainly adapts to drought stress by accumulating large molecular substances such as proline in the root system. After the application of 5-ALA, the drought-induced slow plant growth could be slowly eased, and at the same time, the material accumulation and root formation in the above-ground and underground parts of maize seedlings were promoted, plant growth was shifted from the underground part to the above-ground part, the physiological and biochemical processes of seedling roots were active, and the stability of the plasma membrane was enhanced. In sum, the reasons for the difference in drought resistance between different varieties of the same crop are complex and multi-dimensional. The analysis of root characteristics, metabolic regulation, physiology, and biochemistry can reveal the drought resistance mechanism of maize, broaden breeding ideas, and provide an important basis for drought resistance research.

### 3.2. Transcriptomic Analysis of Drought Responses in Different Inbred Lines Treated with 5-ALA

By comparing the number of differentially expressed genes between different comparison groups, it was found that the number of genes with upregulated expression in TS141 roots was slightly larger than that with downregulated expression under drought stress. After the addition of exogenous 5-ALA treatment, the overall expression of TS141’s differentially genes was abundant, and the number of genes with upregulated expression was much larger than the number with downregulated expression. The number of downregulated genes in the Zheng58 inbred line exceeded that of upregulated genes. After the exogenous application of 5-ALA, differential gene expression and downregulated genes were further increased. The results showed that water deficiency promoted the expression of root genes in response to drought stress, and exogenous 5-ALA accelerated the process and reduced the accumulation of sustained drought stress damage. The number of differentially expressed genes and the difference in upregulated genes between TS141 and Zheng58 indicate that Zheng58 has a more efficient drought tolerance response mechanism and can respond to drought stress quickly and efficiently. This may be the reason why Zheng58 has better drought tolerance than TS141.

GO annotation analysis was performed on the differential genes of different comparison groups, and the results showed that the biological processes mainly enriched after drought were metabolic processes, cellular processes, and monomeric processes in the two inbred lines or under different treatments. The cell groups mainly enriched were divided into cells, cell parts, organelles, and membranes, and the molecular functions mainly enriched were binding and catalytic activities. The results indicated that these pathways were the main pathways of maize seedling root responses to drought. After the application of exogenous 5-ALA, the two inbred lines specifically responded to the activity of organelles and structural molecules, and 5-ALA may regulate the maize root system through these two pathways to achieve drought tolerance.

Differential gene pathway enrichment under exogenous 5-ALA treatment showed that all enrichment pathways were mainly concentrated in the metabolic branch, followed by genetic information processing. Compared with those under normal treatment, the significant enrichment pathways noted by TS141 and Zheng58 under drought stress mainly included MAPK signaling pathways, plant hormone signal transduction, starch and sucrose metabolism, phenylpropionic acid biosynthesis, and plant–pathogen interaction; these pathways were the key pathways for maize seedling root responses to drought. Sucrose is the main form of long-distance transport of plant photosynthetic products, and amino acid phosphorylation promotes sucrose transport in phloem. Compared with those under drought treatment, the pathways of amino acid biosynthesis and starch and sucrose metabolism enrichment were co-noted in the two inbred lines after external application of 5-ALA, indicating that 5-ALA promoted sucrose transport to roots, promoted root growth under drought stress, and increased root osmotic pressure to resist drought. The MAPK signaling pathway is a signal transduction mechanism widely found in eukaryotes. In plants, the MAPK signaling pathway is involved in responses to biotic and abiotic stresses, as well as regulation of plant development and metabolism. Drought stress stimulates the activation of MAPK kinase activator enzyme (MEKK) and other kinases. The activated MAPK kinase can select substrates for phosphorylation modification, thus triggering a series of downstream reactions, such as starting gene expression, regulating enzyme activity, changing cell metabolism to regulate the antioxidant system, regulating membrane permeability, controlling water absorption and transport, and enhancing the resilience of plants to adversity. Plants also regulate endogenous hormones such as ABA, ethylene, and IAA through hormone signal transduction and interact with downstream signal transduction pathways such as cytoplasmic or nuclear protein kinases, transcription factors, gene expression, and metabolic pathways through hormone receptors, thereby regulating biological processes such as plant growth, development, and responses to stress. A MAPK family drought-related gene, *SbMPK14*, was found in sorghum, which reveals a new mechanism whereby MAPK genes regulate drought responses in sorghum [40]. Chen et al.’s [41] research shows that the GhMAP3K62–GhMKK16–GhMPK32 signal cascade can activate the downstream GhEDT1GhNCED3 module and regulate stomatal movement by promoting the accumulation of ABA in cotton leaves, thus enhancing the drought resistance of cotton. Liu et al.’s [42] studies have shown that mitogen-activated protein kinase TaMPK3 inhibits plant responses to ABA by disrupting the stability of TaPYL4 receptors in wheat.

Compared with those in the control, there were 61 differentially expressed genes in the roots of the two inbred lines. After the application of 5-ALA, 143 genes were detected in the two inbred lines, of which 105 were upregulated and 38 were downregulated. Using |log_2_FC| ≥ 2 and *p* ≤ 0.05 as criteria, one downregulated significantly differentially expressed gene and three upregulated significantly differentially expressed genes were screened. These significant genes are mainly involved in histidine metabolism, amino acid biosynthesis, plasma membrane components, secondary active sulfate transmembrane transporter activity, anion reverse transporter activity, and so on. Histidine is an alpha-amino acid containing imidazole rings, which is a biosynthetic precursor of important compounds such as histamine, sarcosine, and geese. In plants, histamine can regulate stomatal opening, promote CTK synthesis and water and oxygen absorption, and enhance plant growth and development. Sarcosine can improve photosynthetic efficiency and increase the sugar contents of plants. Anserine is a multifunctional stress resistance factor, which can improve the resistance of plants to drought, salinity, low temperature, and other stresses. Plants can reduce the synthesis of branched chain amino acids by downregulating the expression of related genes in the amino acid synthesis pathway, thereby increasing the contents of alanine and aspartic acid and promoting plant stress adaptation. Zhang [43] studied the response mechanism of the fluoride content in wild barley leaves to drought stress and found that alanine, valine, and leucine of the pyruvate family were significantly increased in the two groups of wild barley after drought stress, and histidine was also significantly increased. It can be inferred that the glycolytic pathway and carbohydrate catabolic metabolism were enhanced under a drought environment. The response of the starch and sucrose metabolic pathways forces the plant to break down starch to provide energy and carbon sources while promoting the synthesis and accumulation of sucrose to adapt to a water-deficient environment. Li et al. [44] combined transcriptome and metabolome analyses and showed that the metabolic pathways of starch and sucrose are the core metabolic pathways of *Masson pine* seedlings in response to high-temperature and drought interaction stress, while trehalose plays an important regulatory role in the high-temperature and drought interaction stress. Li et al.’s [45] comprehensive transcriptomic and metabolome analyses found that genes and metabolites involved in antioxidant activity, osmotic regulation, benzoic biosynthesis, and flavonoid biosynthesis may regulate the drought adaptation of licorice, and significantly enriched pathways included starch and sucrose metabolism, ascorbic acid metabolism, and uronic acid metabolism, which can balance osmotic potential and reduce oxidative damage. Ion balance inside and outside the cell is achieved by anion transport, and anion transmembrane transporters regulate ion uptake and distribution in plant roots. After the application of 5-ALA, the expression of the *Zm00001d048189* gene, which is related to signal conduction, material transmembrane transport, and anion channel function, was upregulated, and the root tolerance to drought stress was improved. At the same time, 5-ALA promoted the upregulation of the *Zm00001d048444* gene in the two inbred lines in response to the histidine metabolic pathway (ko00340), activated the drought resistance regulatory network, promoted gene expression, induced the metabolism of various amino acids and sugars in the roots of maize seedlings, which were transported to key sites, and promoted the root ion balance, water retention, and fertilizer absorption. The drought tolerance of the plant was improved.

This study discovered that 5-ALA induces drought resistance in maize seedling roots and conducted an in-depth exploration into the mechanisms at both the physiological and transcriptome levels. It was found that adding an appropriate amount of 5-ALA to maize seedlings could accelerate root morphogenesis, significantly increase root biomass, and promote dry matter accumulation and free radical scavenging. This finding provides a new physiological mechanism reference for the study of drought resistance of maize. Transcriptome analysis of maize roots treated with 5-ALA using modern molecular biology techniques revealed the molecular mechanism of alleviating drought stress through complex pathways such as transport and catabolism, signal transduction, amino acid metabolism, lipid metabolism, etc. This provides a theoretical basis for further improving the drought resistance of maize by genetic engineering. This study covers the fields of physiology, plant physiology, molecular biology, etc. and successfully achieved multi-disciplinary cross-application. The fields involved in this research include physiology, plant physiology, molecular biology, etc., which have successfully achieved interdisciplinary applications. Through in-depth exploration from multiple perspectives, the mechanism of 5-ALA-induced drought resistance in maize has been comprehensively revealed. Although this study initially revealed some of the mechanisms of 5-ALA-induced drought resistance, there are still many unknown links and factors that need to be further explored. For example, it can be further studied how 5-ALA enhances the drought resistance of maize by regulating specific gene expression. In addition, this study showed that an appropriate amount of 5-ALA could significantly improve the drought resistance of maize, but different varieties and different growth stages of maize may have different responses to 5-ALA.Therefore, it is necessary to further optimize the application strategy of 5-ALA, such as exploring the optimal concentration and the best use time to improve its application effect. Based on the results of this study, we can try to use gene editing technology to introduce related genes into maize varieties to cultivate new varieties with stronger drought resistance. At the same time, other compounds or technologies with drought resistance potential can also be considered for breeding to develop more effective drought resistance breeding strategies.

## 4. Materials and Methods

### 4.1. Test Materials

The drought-tolerant maize inbred line Zheng58 and drought-sensitive inbred line TS141 were selected by our research group as experimental materials, which were provided by the maize research group of Gansu Agricultural University. PEG and 5-ALA were selected based on the optimal stress and slow-release concentrations previously screened by the research group, with a mass fraction of 15% PEG6000 and 25 mg/L 5-ALA, respectively.

### 4.2. Test Methods

Plump and uniform-sized maize seed were selected and disinfected with 2% sodium hypochlorite solution for 10 min, rinsed with distilled water 3 times, and placed in distilled water until the seeds were fully absorbed. Seeds were seeded into a flowerpot (diameter 10 cm, height 11 cm) filled with vermiculite and placed in an artificial climate incubator (RTOP-310Y, Zhejiang Top Instrument Co., Ltd., Hangzhou, China) with a day/night temperature of 25/20 °C, light intensity of 600 μmol/s/m^2^, and relative humidity of 60–80%. After the seedlings grew to three leaves, 20 mL of distilled water was poured every 2 d as the normal control, drought stress was executed with 15%PEG solution, and 25 mg/L5-ALA + 15% PEG mixed solution was used as an exogenous sustained release treatment (AP). Root growth parameters were measured on the 7th day of treatment with 8 plants in each pot and 5 plants in each pot [46]. The samples were stored at −80 °C for the determination of physiological and biochemical parameters and transcriptome sequencing.

### 4.3. Measurement Items and Methods

#### 4.3.1. Measurement of Growth Parameters and Physiological and Biochemical Indices

Plant growth parameter determination: Peng et al.’s [47] method was referenced to determine seedling length, seedling fresh weight, main radicle length, and root fresh length and other phenotypic traits. The root surface area, total root length, and average root diameter were measured by an automatic root scanning analyzer (IN-GX02).

Determination of relative water content: the RWC of roots was determined by the drying method [48].

Plant biomass and root/shoot ratio: 10 seedlings were taken from each treatment and divided into two parts, the above-ground part and the root part. The seedlings were defoliated at 110 °C for 10 min. The biomass was weighed after drying at 75 °C to a constant amount [49].

Root/shoot ratio = Root biomass/shoot biomass.

Plant biomass = Shoot biomass + root biomass [49].

Physiological parameter determination: proline, MDA, and REC were measured according to the Experiments Guidelines of Plant Physiology [50]. POD, CAT, and SOD were determined by the sulfosalicylic acid method, guaiacol method, and nitrogen blue tetrazole method, respectively [51].

#### 4.3.2. Construction and Sequencing of the RNA-Seq Library

Transcriptome sequencing was performed using the Illumina platform (www.biomarker.com.cn, accessed on 2 February 2022). The total RNA of 18 root samples stored at −80 °C was isolated and purified by a TRIzol (Invitrogen, Carlsbad, CA, USA) kit. The RNA-seq library was constructed with the total RNA of each treated sample whose purity met the standard of library construction. The library was completed by Beijing Biomarker Technologies Co., Ltd. (Beijing, China). The effective concentration of the constructed library was accurately quantified by qPCR to ensure the quality of the library.

#### 4.3.3. Quality Assessment of Sequencing Results

Clean reads were obtained by filtering the raw reads obtained by sequencing, and subsequent analysis was based on clean reads. The filtering software SOAPnuke 1.X independently developed by BTU was used for filtering, and clean reads in FASTQ format were obtained and analyzed statistically. Clean reads of the sample were compared with the fourth edition of the B73 maize reference genome (GCF_000005005.2_B73_RefGen_v4) using the software HISAT 2.0.4. Reads after comparison were assembled using StringTie v2.2.3. The expression levels of genes were calculated by FPKM [52] (fragments per kilobase of transcript per million fragments mapped).

#### 4.3.4. Analysis of Differentially Expressed Genes

The DESeq2 tool (v1.40.2) was used for differential gene analysis, and the screening criteria were fold change ≥ 2 and FDR < 0.01 [53]. The differentially expressed genes were compared with the KEGG database and GO annotations to determine their biochemical and metabolic pathways and signal transduction pathways. Comparing the background genome with pathway as a unit, hypergeometric tests were used to identify the significantly enriched pathways in differentially expressed genes to determine the most important biochemical metabolic pathways and signal transduction pathways involved in DEGs.

#### 4.3.5. qRT-PCR Verification of Differentially Expressed Genes

The established RNA-seq library was reverse transcribed using the M-MLV II Reverse Transcription Kit (Eric Bio, Guangzhou, China) to obtain cDNA as a template for PCR amplification. We used the Illumina Eco real-time fluorescence quantitative PCR instrument and SYBR Green Pro Taq HS premixed qPCR kit (Eric Bio, Guangzhou, China) for amplification. The relative expression level was calculated using the 2^−ΔΔCt^ method [54], and the primers are shown in Table 3.

### 4.4. Statistics Analysis of Data

Microsoft Excel 2019 software was used for statistical analysis and mapping of test data, IBM SPSS Statistics26 software was used for data analysis and evaluation, and WinRHIZO v2.0 software was used to analyze root data.

## 5. Conclusions

Our research findings suggest that the introduction of 25 mg/L 5-ALA under drought stress conditions can expedite root morphogenesis in maize seedlings. It not only promotes the accumulation of dry matter and free radical scavenging but also alleviates drought stress in maize roots through a range of complex biochemical pathways. These include transport and decomposition metabolism, signal transduction, amino acid metabolism, lipid metabolism, carbon metabolism, translation, and phenylpropanoid biosynthesis. By regulating the osmotic system, photosynthetic system, and peroxide system in maize, 5-ALA enhances the drought tolerance of the crop. Our study offers a valuable reference for understanding the drought tolerance mechanisms in maize and for utilizing exogenous hormones as a strategy to mitigate drought stress in maize.

## Figures and Tables

**Figure 1 ijms-25-12963-f001:**
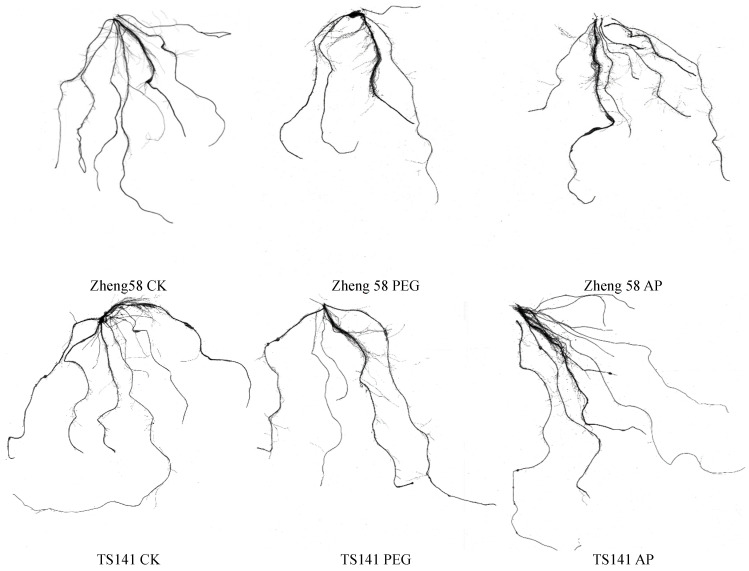
Root scans of maize inbred seedlings under different treatments. CK: distilled water treatment; PEG: 15% PEG treatment; AP: 25 mg/L5-ALA + 15%PEG treatment.

**Figure 2 ijms-25-12963-f002:**
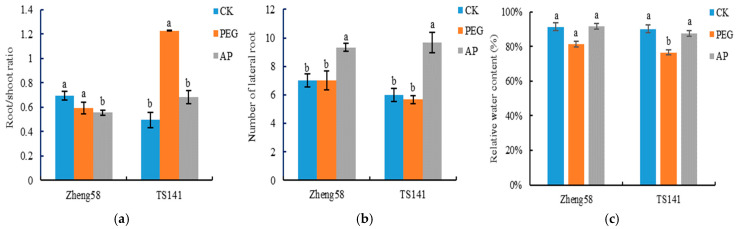
Different lowercase letters represent the same inbred line with significant differences under different treatments (*p* < 0.05). CK: distilled water treatment; PEG: 15% PEG treatment; AP: 25 mg/L5-ALA + 15%PEG treatment. (**a**) Root/shoot ratio; (**b**) Number of lateral roots; (**c**) Relative water content.

**Figure 3 ijms-25-12963-f003:**
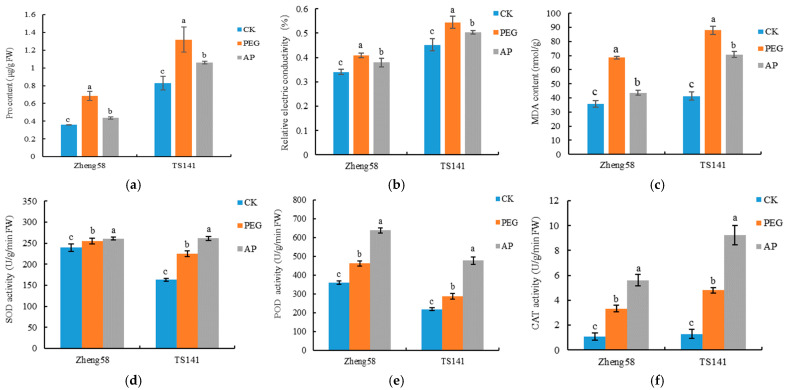
Different lowercase letters represent the same inbred line with significant difference under different treatments (*p* < 0.05). CK: distilled water treatment; PEG: 15% PEG treatment; AP: 25 mg/L5-ALA + 15% PEG treatment. (**a**) Proline content; (**b**) Relative electric conductivity; (**c**) MDA content; (**d**) SOD activity; (**e**) POD activity; (**f**) CAT activity.

**Figure 4 ijms-25-12963-f004:**
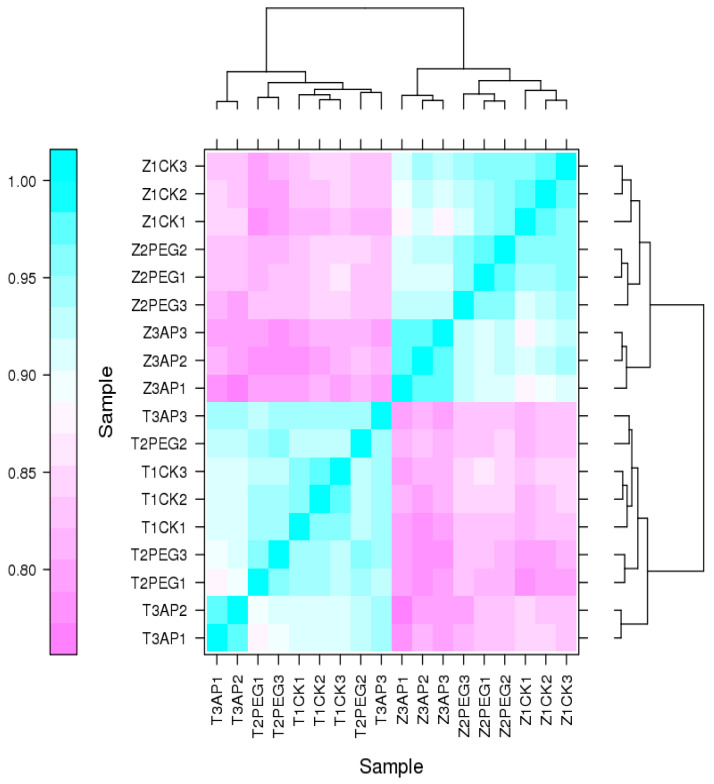
Heatmap of the expression quantity correlation between two samples. T-CK: Distilled water treatment TS141; T-PEG: 15% PEG processing TS141; T-AP: 25 mg/L 5-ALA + 15% PEG to treat TS141; Z-CK: Distillation water treatment Zheng58; Z-PEG: 15% PEG treatment Zheng58; Z-AP: 25 mg/L 5-ALA + 15% PEG treatment Zheng58; The number after the sample number indicates that the same treatment is repeated.

**Figure 5 ijms-25-12963-f005:**
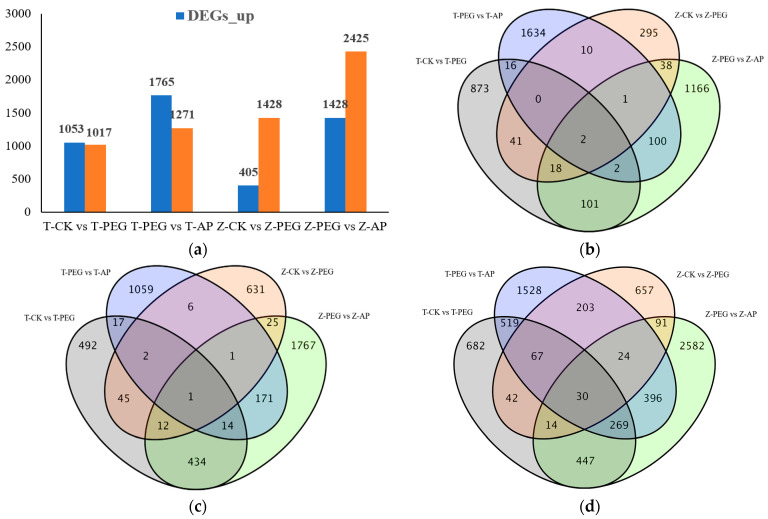
Upregulated and downregulated DEG numbers and Venn analysis between different treatments. (**a**) Downregulated distribution of DEG numbers in different treatment groups; (**b**) Venn analysis of expressed genes in different treatment groups; (**c**) Venn diagram analysis of upregulated DEG numbers in different treatment groups; (**d**) Venn analysis of downregulated DEG numbers in different treatment groups.

**Figure 6 ijms-25-12963-f006:**
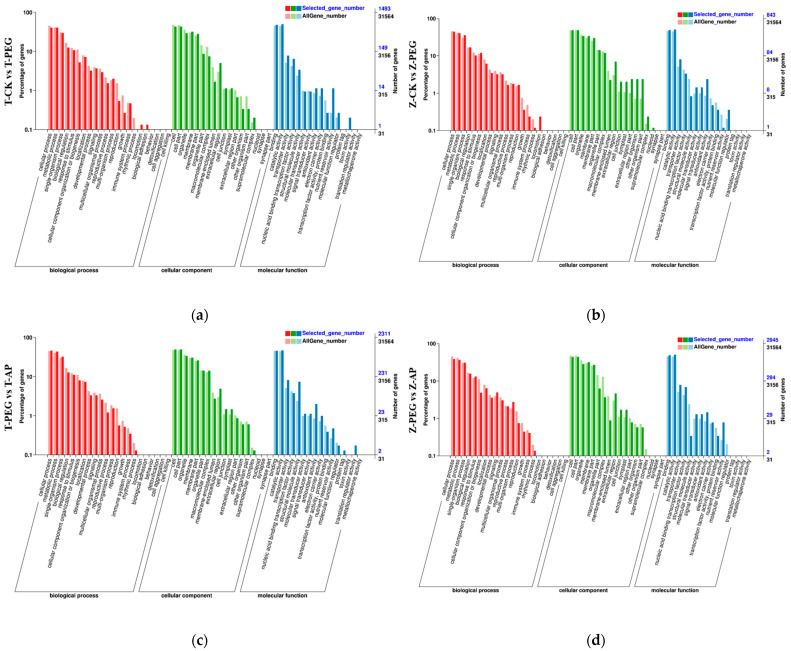
GO annotations of inbred lines TS141 and Zheng58 under different treatments. (**a**) GO annotations for inbred line TS141 under normal treatment and drought stress; (**b**) GO annotations for inbred line Zheng58 under normal treatment and drought stress; (**c**) GO annotation for inbred line TS141 under 5-ALA application; (**d**) GO annotation for inbred line Zheng58 under 5-ALA application.

**Figure 7 ijms-25-12963-f007:**
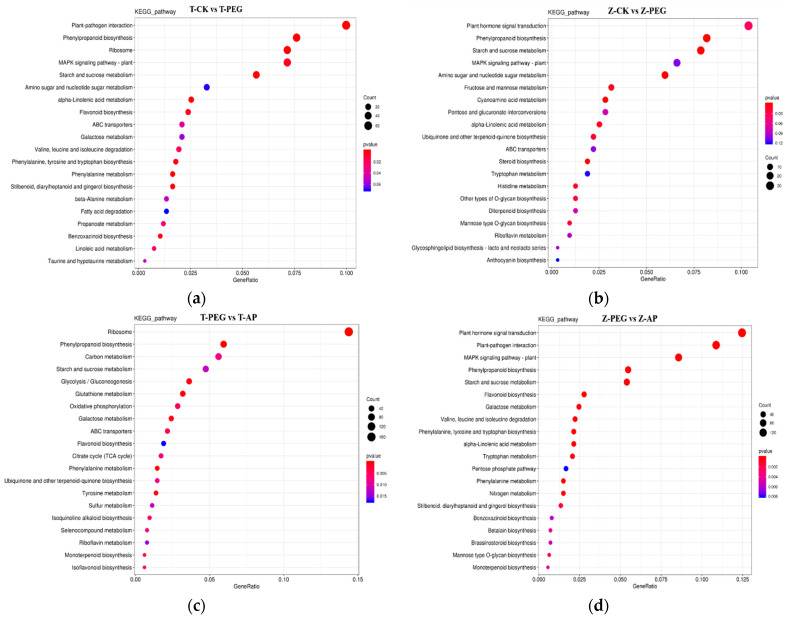
KEGG metabolic pathway analysis of TS141 and Zheng58 under different treatments. (**a**) KEGG metabolic pathways of TS141 under normal treatment and drought stress; (**b**) KEGG metabolic pathways of Zheng58 under normal treatment and drought stress; (**c**) KEGG metabolic pathway of TS141 under external application of 5-ALA; (**d**) KEGG metabolic pathway of Zheng58 under 5-ALA application.

**Figure 8 ijms-25-12963-f008:**
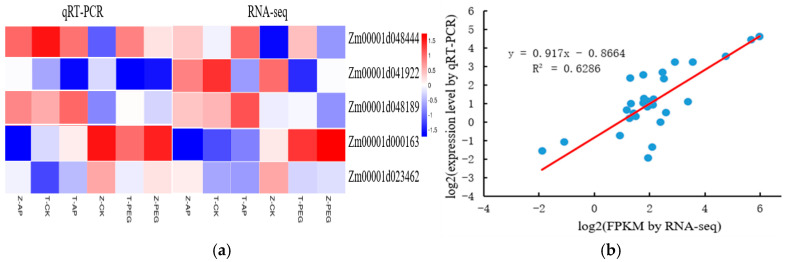
The expression patterns of 5 selected genes identified by RNA-seq was verified by qRT-PCR. (**a**) Heat map showing the expression changes (logy-fold change) in response to the Z-AP, T-CK, T-AP, Z-CK, T-PEG, and Z-PEG treatments for each candidate gene as measured by RNA-seq and qRT-PCR; (**b**) Scatter plot showing the changes in the expression (logy-fold change) of selected genes based on RNA-seq via qRT-PCR. Gene expression levels are indicated by colored bars.

**Figure 9 ijms-25-12963-f009:**
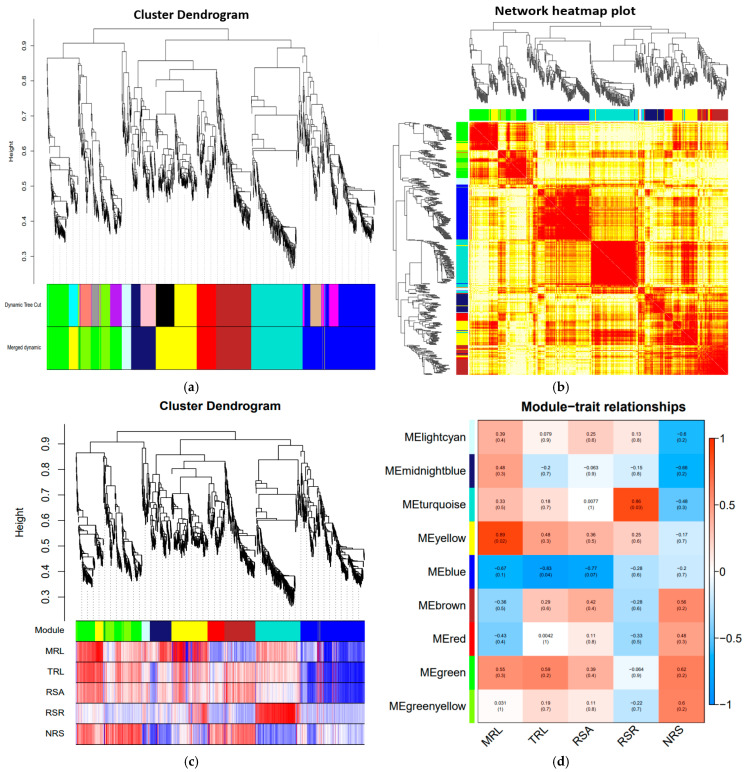
Module construction based on WGCNA. (**a**) Gene network module; (**b**) Gene co-expression network heat map; (**c**) Gene phylogenetic tree and trait correlation heat map; (**d**) Heatmap of correlations between modules and traits. The closer the correlation is to the absolute value of 1, the more relevant the trait is to the gene of the module.

**Figure 10 ijms-25-12963-f010:**
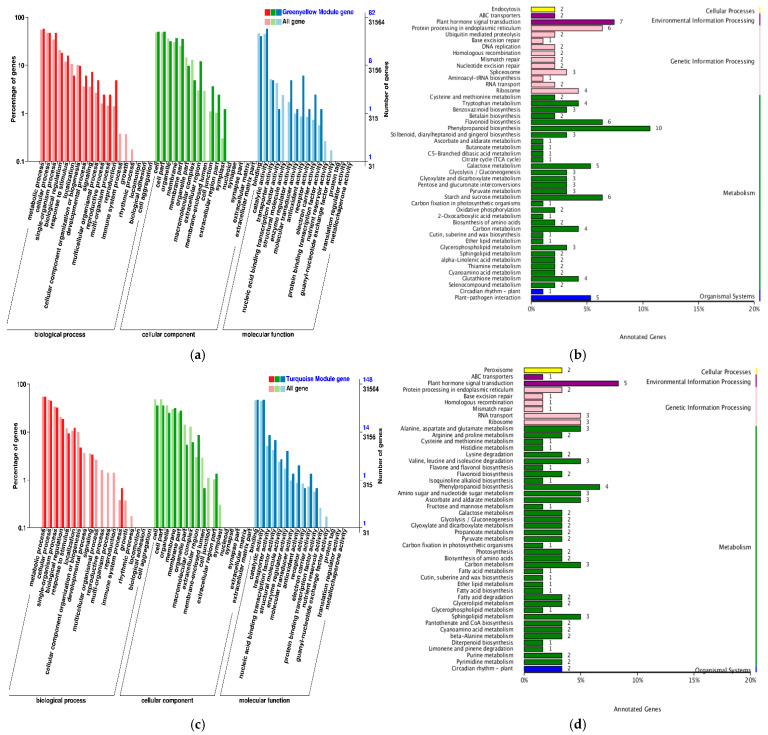
Functional analysis of genes in the blue and turquoise modules. (**a**) GO enrichment analysis in the yellow module; (**b**) KEGG enrichment analysis in the yellow module; (**c**) GO enrichment analysis in the turquoise module; (**d**) KEGG enrichment analysis in the turquoise module.

**Figure 11 ijms-25-12963-f011:**
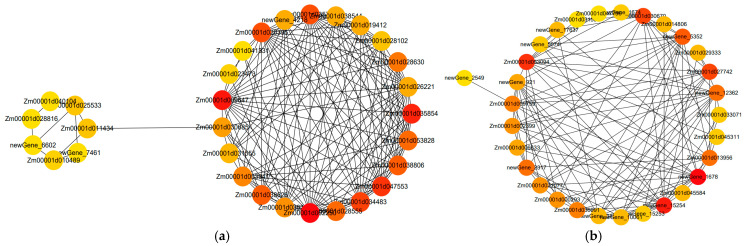
Co-expression regulatory network analysis of the blue module. Red represents hub genes. (**a**) Network interaction analysis of hub genes in the yellow module; (**b**) Network interaction analysis of hub genes in the turquoise module. The color gradients of the dots represent high or low soft thresholds of connectivity, with a redder dot color representing a higher soft threshold of connectivity.

**Table 1 ijms-25-12963-t001:** Changes in the growth of inbred maize seedlings under different treatments.

Varieties	Treatments	Seedling Length (cm)	Seedling Fresh Weight (g)	Shoot Dry Weight (g)	Plant Biomass (g)
Zheng58	CK	33.10 ± 0.70 a	0.82 ± 0.005 b	0.08 ± 0.003 a	0.14 ± 0.002 a
	PEG	27.30 ± 0.50 b	0.74 ± 0.185 b	0.07 ± 0.013 a	0.12 ± 0.016 a
	AP	32.80 ± 0.10 a	1.16 ± 0.040 a	0.08 ± 0.004 a	0.12 ± 0.003 a
TS141	CK	31.15 ± 0.75 a	0.90 ± 0.035 b	0.10 ± 0.009 a	0.14 ± 0.003 a
	PEG	27.75 ± 3.65 a	0.67 ± 0.010 c	0.05 ± 0.003 b	0.08 ± 0.004 b
	AP	29.80 ± 4.30 a	1.17 ± 0.030 a	0.09 ± 0.003 a	0.14 ± 0.014 a

Different lowercase letters, a, b, and c, represent significant differences (*p* < 0.05) for the same inbred line under different treatments. CK: distilled water treatment; PEG: 15% PEG treatment; AP: 25 mg/L5-ALA + 15% PEG treatment.

**Table 2 ijms-25-12963-t002:** Changes in the root growth of maize inbred seedlings under different treatments.

Cultivar	Treatment	Root Dry Weight (g)	Root Average Diameter (cm)	Main Root Length (cm)	Total Root Length (cm)	Root Surface Area (cm^2^)
Zheng58	CK	0.056 ± 0.003 a	0.46 ± 0.016 a	22.80 ± 4.78 a	386.59 ± 16.02b	56.31 ± 2.64 b
	PEG	0.035 ± 0.003 a	0.46 ± 0.005 a	15.33 ± 2.37 a	358.48 ± 20.53b	52.00 ± 4.04 b
	AP	0.044 ± 0.001 b	0.48 ± 0.008 a	22.30 ± 2.01 a	468.43 ± 15.41a	70.02 ± 2.31 a
TS141	CK	0.046 ± 0.006 b	0.47 ± 0.015 a	33.87 ± 1.53 a	431.08 ± 6.91b	66.35 ± 1.83 ab
	PEG	0.057 ± 0.002 a	0.44 ± 0.007 a	26.57 ± 2.54 b	422.39 ± 6.72b	58.50 ± 2.01 b
	AP	0.058 ± 0.016 a	0.47 ± 0.004 a	35.47 ± 2.07 a	492.83 ± 8.03a	68.80 ± 2.30 a

Different lowercase letters, a, b, and c, represent significant differences (*p* < 0.05) for the same inbred line under different treatments. CK: distilled water treatment; PEG: 15% PEG treatment; AP: 25 mg/L5-ALA + 15% PEG treatment.

**Table 3 ijms-25-12963-t003:** qRT-PCR primers.

Gene ID	Forward Primer	Reverse Primer
Action	TGAAACCTTCGAATGCCCAG	GATTGGAACCGTGTGGCTCA
Zm00001d048444	GGACATGGCGGTGGTGATGAAG	AGCAGCTCGATCTCCTCCTTGG
Zm00001d041922	GCTTGCCGTGGCGTTCTGG	GGAGGAGGTGTGACGACTGGAG
Zm00001d048189	GCTGCCATCACCATCGCTCTTC	CAACCAGAAGGAATGCCAGGAAGG
Zm00001d048894	CTGCCTGTGCCGTAGCGT	TGTGACGACTGGAGGTGGC

## Data Availability

The datasets and materials used and/or analyzed during the current study are available from the corresponding author upon reasonable request.

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
