# Peer review of "5-Aminolevulinic Acid (5-ALA)-Induced Drought Resistance in Maize Seedling Root at Physiological and Transcriptomic Levels"

_ijms, 2024, doi:10.3390/ijms252312963_

Round 1
Reviewer 1 Report
Comments and Suggestions for Authors
Comments and Suggestions for Authors
Dear Author
I have an honor to review the manuscript entitled “5-ALA Induced Drought Resistance of Contrasting Drought-responsive Genotypes of Maize Seedling Root at Physiological and Transcriptomic Levels” a research article submitted to the MDPI Journal, IJMS. Authors of this manuscript induced drought by 5-ALA and analyzed its effects on maize seedlings root through a series of molecular research. Further, identified differentially expressed genes by transcriptome analysis and confirmed its validity by RT-PCR. Also performed a series of bioinformatic analysis of the gens. Overall, the experiments they performed are well and the results are convincing. Thus, the presented results take up an important topic consistent with the profile of the Journal.
-However, even, manuscript is well organized and well described of the conception; I have some suggestions, which might improve the manuscript to make important to the wider audience.
-Used English is very difficult for clarity off the research. It should be improved throughout the text.
-Most suggestions I have mentioned in the main text pdf file. Please check
Title: -Good organization with results order.
Here, 1st part; 5-ALA induced drought resistant at physiological level-------it is ok
But, 2nd part; 5-ALA induced drought resistant at transcriptomic level------does it give any sense?. Write easy understanding and meaningful title
Abstract:
- Write full form at their first time appearance. Follow throughout the text
-L14-15; It is vague. Write specific identification. What type morphogenesis, what substances accumulates? Need clear information in the abstract
-Abstract should be rewritten with core results obtained rather than theoretical basis of the research.
1. Introduction
-L38-42; Unnecessarily elongated, redundantly applied, misleading connection between subjects. This description distracts the continuous flow of the reading. Make easy sentences to get main information easily.
-Many times used “maize seedling roots”---what about only root or maize root?
-L104-108; Many time used “under drought stress” redundant.
-The aim of the study should be underlined precisely and simultaneously and highlight why this study is important. Rationale to be elucidated for the purpose of the study.
Results:
-So many time used “drought-tolerant inbred line” “drought-sensitive inbred line” . You need to reduce the frequency of using similar term repeatedly.
-Title 2.1 and 2.2 are same; how comes? It is not the way of writing scientific paper.
-You have same problem in many places. Change all with proper indicating title.
-Also, Table 1 and Table 2; title are same.
-Some parts are not like your results. You can set them in introductory or discussion. Like L156-159.
-L261; The results indicated that the inbred line TS141 was more susceptible to drought stress and had active gene expression, while the inbred line Zheng 58 adapted to drought stress by downregulating gene expression.
This statement is ambiguous. Just say up or down regulated genes statistics among genotypes
-Title 2.5.4.1 and 2.5.4.2; what is basic difference these two? Same meaning
- 2.6, 2.7; same
-2.7.2, 2.7.3; same
Percent match: 32%
Should be below 20%

The English could be improved to more clearly express the research.
Author Response
|
3. Point-by-point response to Comments and Suggestions for Authors
Title: -Good organization with results order. Comments 1: Here, 1st part; 5-ALA induced drought resistant at physiological level-------it is ok But, 2nd part; 5-ALA induced drought resistant at transcriptomic level------does it give any sense?. Write easy understanding and meaningful title Response 1: Thank you for pointing this out. We agree with this comment. We have revised the title according to your suggestion. (Line2-Line3) Abstract: Comments 2: Write full form at their first time appearance. Follow throughout the text Response 2: Thank you for pointing this out. We agree with this comment. We have revised the title according to your suggestions. (Line2, Line181, Line196,) Comments 3: L14-15; It is vague. Write specific identification. What type morphogenesis, what substances accumulates? Need clear information in the abstract Response 3: Thank you for pointing this out. We agree with this comment. We have revised this content according to your suggestions. (Line13- Line14) Comments 4: Abstract should be rewritten with core results obtained rather than theoretical basis of the research. Response 4: Thank you for pointing this out. We agree with this comment. We have revised abstract according to your suggestions. (Line10- Line26)
1. Introduction Comments 5: L38-42; Unnecessarily elongated, redundantly applied, misleading connection between subjects. This description distracts the continuous flow of the reading. Make easy sentences to get main information easily. Response 5: Thank you for pointing this out. We agree with this comment. We have revised this content according to your suggestions. (Line53- Line58) Comments 6: Many times used “maize seedling roots” ---what about only root or maize root? Response 6: Thank you for pointing this out. What we are referring to is the roots of corn seedlings during the seedling stage. (Line53- Line58) Comments 7: L104-108; Many time used “under drought stress” redundant. Response 7: Thank you for pointing this out. We agree with this comment. We have revised this content according to your suggestions. Comments 8: The aim of the study should be underlined precisely and simultaneously and highlight why this study is important. Rationale to be elucidated for the purpose of the study. Response 8: Thank you for pointing this out. We agree with this comment. We have supplemented this content according to your suggestions. (Line94- Line111)
Results: Comments 9: So many time used “drought-tolerant inbred line” “drought-sensitive inbred line” . You need to reduce the frequency of using similar term repeatedly. Response 9: Thank you for pointing this out. We agree with this comment. We have supplemented this content according to your suggestions. Comments 10: Title 2.1 and 2.2 are same; how comes? It is not the way of writing scientific paper. Response 10: Thank you for pointing this out. Sorry, the title of 2.2 was written incorrectly by us and we have made the necessary changes. (Line130) Comments 11: You have same problem in many places. Change all with proper indicating title. Also, Table 1 and Table 2; title are same. Response 11: Thank you for pointing this out. Sorry, the title of Table 2 was written incorrectly by us and we have made the necessary changes. (Line146) Comments 12: Some parts are not like your results. You can set them in introductory or discussion. Like L156-159. Response 12: Thank you for pointing this out. We have made modifications to the results section based on your feedback. (Line427- Line428, Line431- Line436, Line449- Line452, Line466- Line467, Line477- Line479, Line500- Line503) Comments 13: L261; The results indicated that the inbred line TS141 was more susceptible to drought stress and had active gene expression, while the inbred line Zheng 58 adapted to drought stress by downregulating gene expression. This statement is ambiguous. Just say up or down regulated genes statistics among genotypes Response 13: Thank you for pointing this out. We agree with this comment. We have deleted this content according to your suggestions. (Line240- Line246) Comments 14: Title 2.5.4.1 and 2.5.4.2; what is basic difference these two? Same meaning Response 14: Thank you for pointing this out. Title 2.5.4.1 is a GO analysis of two inbred lines without the addition of 5-ALA under drought stress. Title 2.5.4.2 is the GO analysis of two inbred lines after adding 5-ALA treatment. (Line296) Comments 15: 2.6, 2.7; same Response 15: Thank you for pointing this out. Sorry, the title of 2.7 was written incorrectly by us and we have made the necessary changes. (Line341) Comments 16: 2.7.2, 2.7.3; same Response 16: Thank you for pointing this out. Thank you for pointing this out. Sorry, the title of 2.7.2 was written incorrectly by us and we have made the necessary changes. (Line358) Comments 17: Percent match: 32% Should be below 20% Response 17: Thank you for pointing this out. We have modified the percent match and it should be below 20%.
|

Reviewer 2 Report
Comments and Suggestions for Authors
Journal IJMS (ISSN 1422-0067)
Manuscript ID ijms-3323326
Type Article
Title 5-ALA Induced Drought Resistance of Contrasting Drought-responsive Genotypes of Maize Seedling Root at Physiological and Transcriptomic Levels
Authors Ya qiong Shi , Zi hao Jin , Jing yi Wang , Guang kuo Zhou , Fang Wang * , Yun ling Peng
Section Molecular Plant Sciences
Dear Editor
I am submitting a review of the manuscript.
I have included below comments on the text.
With kind regards
Reviewer
TITLE
1.
Please elaborate on the abbreviation.
KEY WORDS
2.
Please eliminate terms that appear in the title of the manuscript.
INTRODUCTION
l. 28-42.
3.
Please provide in English the name of the species instead of the genus ‘Maize’
4.
Please, for the given: country, area, region, underline maize area, yield, economic, market demand, etc.
5.
Please state which regions and conditions why “…maize in many world regions…”
6.
Please indicate other factors limiting maize productivity, disruption of metabolic processes at cellular level.
7.
What mechanisms at the organismal and cellular level determine drought resistance (osmoregulation, appropriate cations, small-molecule substances, membrane integrity, etc.) with reference to the manuscript topic.
8.
Please complete what ‘artificial regulation to minimize the damage caused by drought stress.
l. 43-53.
9.
Please complete the substantive explanation of the following processes so that they are understandable to any reader and the cited publication has the process in question in the title of the manuscript instead of “enumeration”:
(ii) osmotic regulation,
(ii) hormone balance,
(iii) accumulation of some essential ions and mineral elements,
(iv) water use efficiency
(v) antioxidant defence system
10.
l. 45 - 48. Please complete these regulators and state their mechanism in terms of the subject matter of the manuscript.
11.
l.49 - 53. Please describe the processes listed in relation to the topic of the manuscript, citing recent publications having the metabolic process in question in the title....
12.
l. 45 - 53. No publication citation, but after rewording, please cite according to comment number 11.
l. 54 - 109.
13.
L. 54 – 109. Covers 782 words, please separate thematic paragraphs in this text.
14.
l. 54 – 64. Please refer to the theme of the manuscript.
15.
l. 64 - 67. Please refer to the subject of the manuscript.
16.
l. 67-69. Complete what enzymatic antioxidants and non-enzymatic antioxidants work with reference to the topic of the manuscript.
17.
Please complete the factual information regarding clade monocotyledonous plants. However, the quoted plant genera: “rape [8], cucumber [9], and sunflower [10], ...coral seedlings... strawberry, ...” are from taxonomically distant botanical families.
18.
l. 69-72. please explain these metabolic processes with reference to the theme of the manuscript
“...improving the photosynthetic electron transfer efficiency and improving the drought-resistant ability of plants...”.
19.
l. 72-78. Please complete the explanation of these metabolic processes at the cellular level with reference to the manuscript topic.
20.
l. 78-80. please check the subject of the manuscript and rewrite the text.
21.
l. 80-81. The authors enumerate the effect, but the researchers and readers are interested in the metabolic, biochemical process at the cellular level. Please, instead of enumerating the effect, complete the explanation of the piercing metabolic processes in close reference to the manuscript topic.
22.
l. 81-83. Peony belongs to the family Paeoniaceae, a distant family in the taxonomic hierarchy from Zea mays, Please check what the subject of the manuscroipt “Maize Seedling Root” refers to and reword the text.
23.
l. 99-109. Please reinforce the rationale for the manuscript topic undertaken.
24.
Please formulate the specific aim of the paper.
RESULTS
25.
Table 1 and 2. Please note the figures in the table, whether the results can be uniformly placed to two decimal places.
26.
Results are not the place to cite literature items: l. 154, 155, 171, 191-193, 212, 239-243. Not here.
27.
Figure 6-11 The posted information is illegible.
DISCUSSION
28.
Please complete the innovative achievement of the research carried out.
29.
Please complete the perspective of the research in the future .
MATERIALS AND METHODS
30.
Please note whether, for each step, the relevant literature item is quoted confirming the compatibility of the method undertaken.
CONCLUSION
31.
Please complete the specific conclusions of your research.
REFENCES
32.
Please edit each item of literature according to the guidelines for authors, for example
- Latin names should be written in italics l. 768, 779, 784, 808, 874 etc.
- please write journal titles correctly
(i) once are all words in capital letters e.g. l. 755
(ii) the second time only the first word in capital letters l. 771, 844.
(iii) the third time an abbreviation is given e.g. l. , 803, 874.
It is impossible to list them all.

Author Response
TITLE
Comments 1: Please elaborate on the abbreviation.
Response 1: Thank you for pointing this out. We agree with this comment. Therefore, we have provided a detailed explanation of the abbreviation. (Line2)
KEY WORDS
Comments 2: Please eliminate terms that appear in the title of the manuscript.
Response 2: Thank you for pointing this out. We have made modifications to the keywords in the paper. (Line27- Line28)
INTRODUCTION
- 28-42.
Comments 3: Please provide in English the name of the species instead of the genus ‘Maize’
Response 3: Thank you for pointing this out. Maize refers to the English name of a species, we have searched a lot of papers and it is written like this. I’m sorry, and I didn't understand what you meant.
Comments 4: Please, for the given: country, area, region, underline maize area, yield, economic, market demand, etc.
Response 4: Thank you for pointing this out. We have made revisions to the preface section based on your feedback and added relevant content. (Line32-Line34)
Comments 5: Please state which regions and conditions why “…maize in many world regions…”
Response 5: Thank you for pointing this out. We have made revisions to the preface section based on your feedback and added relevant content. (Line32-Line34)
Comments 6: Please indicate other factors limiting maize productivity, disruption of metabolic processes at cellular level.
Response 6: Thank you for pointing this out. We have made revisions to the preface section based on your feedback and added relevant content. (Line36-Line44)
Comments 7: What mechanisms at the organismal and cellular level determine drought resistance (osmoregulation, appropriate cations, small-molecule substances, membrane integrity, etc.) with reference to the manuscript topic.
Response 7: Thank you for pointing this out. We have made revisions to the preface section based on your feedback. (Line46-Line53)
Comments 8: Please complete what ‘artificial regulation to minimize the damage caused by drought stress.
Response 8: Thank you for pointing this out. We have made revisions to the preface section based on your feedback. (Line53-Line58)
- 43-53.
Comments 9: Please complete the substantive explanation of the following processes so that they are understandable to any reader and the cited publication has the process in question in the title of the manuscript instead of “enumeration”:
(ii) osmotic regulation,
(ii) hormone balance,
(iii) accumulation of some essential ions and mineral elements,
(iv) water use efficiency
(v) antioxidant defence system
Response 9: Thank you for pointing this out. We have made revisions to the preface section based on your feedback and this related content has been deleted. (Line31-Line58)
Comments 10: 45 - 48. Please complete these regulators and state their mechanism in terms of the subject matter of the manuscript.
Response 10: Thank you for pointing this out. We have made revisions to the preface section based on your feedback and this related content has been deleted. (Line31-Line58)
Comments 11:49 - 53. Please describe the processes listed in relation to the topic of the manuscript, citing recent publications having the metabolic process in question in the title....
Response 11: Thank you for pointing this out. We have rewritten this section based on the theme of the article. (Line59-Line93)
Comments 12: 45 - 53. No publication citation, but after rewording, please cite according to comment number 11.
Response 12: Thank you for pointing this out. We have rewritten this section based on the theme of the article. (Line59-Line936)
- 54 - 109.
Comments 13: 54 – 109. Covers 782 words, please separate thematic paragraphs in this text.
Response 13: Thank you for pointing this out. We have rewritten this section. (Line59-Line93)
Comments 14: 54 – 64. Please refer to the theme of the manuscript.
Response 14: Thank you for pointing this out. We have reviewed the literature again and made revisions to this section. (Line59-Line93)
Comments 15: 64 - 67. Please refer to the subject of the manuscript.
Response 15: Thank you for pointing this out. We have reviewed the literature again and made revisions to this section. (Line59-Line93)
Comments 16: 67-69. Complete what enzymatic antioxidants and non-enzymatic antioxidants work with reference to the topic of the manuscript.
Response 16: Thank you for pointing this out. We have reviewed the literature again and made revisions to this section. (Line59-Line93)
Comments 17: Please complete the factual information regarding clade monocotyledonous plants. However, the quoted plant genera: “rape [8], cucumber [9], and sunflower [10], ...coral seedlings... strawberry, ...” are from taxonomically distant botanical families.
Response 17: Thank you for pointing this out. We have reviewed the literature again and made revisions to this section. (Line59-Line93)
Comments 18: 69-72. please explain these metabolic processes with reference to the theme of the manuscript
“...improving the photosynthetic electron transfer efficiency and improving the drought-resistant ability of plants...”.
Response 18: Thank you for pointing this out. We have made revisions to the preface section based on your feedback. (Line59-Line93)
Comments 19: 72-78. Please complete the explanation of these metabolic processes at the cellular level with reference to the manuscript topic.
Response 19: Thank you for pointing this out. We have made revisions to the preface section based on your feedback and added relevant content. (Line59-Line93)
Comments 20: 78-80. please check the subject of the manuscript and rewrite the text.
Response 20: Thank you for pointing this out. We have rewritten this section. (Line59-Line93)
Comments 21: 80-81. The authors enumerate the effect, but the researchers and readers are interested in the metabolic, biochemical process at the cellular level. Please, instead of enumerating the effect, complete the explanation of the piercing metabolic processes in close reference to the manuscript topic.
Response 21: Thank you for pointing this out. We have made revisions to the preface section based on your feedback and added relevant content. (Line68-Line85)
Comments 22: 81-83. Peony belongs to the family Paeoniaceae, a distant family in the taxonomic hierarchy from Zea mays, Please check what the subject of the manuscroipt “Maize Seedling Root” refers to and reword the text.
Response 22: Thank you for pointing this out. We have modified this part of the content. (Line68-Line91)
Comments 23: 99-109. Please reinforce the rationale for the manuscript topic undertaken.
Response 23: Thank you for pointing this out. We have supplemented the purpose of the research. (Line94-Line99)
Comments 24: Please formulate the specific aim of the paper.
Response 24: Thank you for pointing this out. We have made revisions to the preface section based on your feedback and added relevant content. (Line94-Line111)
RESULTS
Comments 25: Table 1 and 2. Please note the figures in the table, whether the results can be uniformly placed to two decimal places.
Response 25: Thank you for pointing this out. We placed the result data uniformly to two decimal places. (Line126, Line146)
Comments 26: Results are not the place to cite literature items: l. 154, 155, 171, 191-193, 212, 239-243. Not here.
Response 26: Thank you for pointing this out. We agree with this comment. We have placed this part of the results in the discussion section later. (Line427- Line428, Line431- Line436, Line449- Line452, Line466- Line467, Line477- Line479, Line500- Line503)
Comments 27: Figure 6-11 The posted information is illegible.
Response 27: Thank you for pointing this out. We have adjusted the clarity of the images. (Line172, Line203, Line227, Line259, Line290, Line323, Line335, Line353, Line377, Line405)
DISCUSSION
Comments 28: Please complete the innovative achievement of the research carried out.
Response 28: Thank you for pointing this out. We supplemented the innovative achievements of the research conducted during the discussion. (Line640- Line656)
Comments 29: Please complete the perspective of the research in the future.
Response 29: Thank you for pointing this out. We added prospects for future research during the discussion. (Line657- Line668)
MATERIALS AND METHODS
Comments 30: Please note whether, for each step, the relevant literature item is quoted confirming the compatibility of the method undertaken.
Response 30: Thank you for pointing this out. We agree with this comment. We have added references for each step. (Line687, Line698, Line702, Line703, Line704)
CONCLUSION
Comments 31: Please complete the specific conclusions of your research.
Response 31: Thank you for pointing this out. We have rewritten the content of the conclusion section. (Line744- Line753)
REFENCES
Comments 32:
Please edit each item of literature according to the guidelines for authors, for example
- Latin names should be written in italics l. 768, 779, 784, 808, 874 etc.
- please write journal titles correctly
(i) once are all words in capital letters e.g. l. 755
(ii) the second time only the first word in capital letters l. 771, 844.
(iii) the third time an abbreviation is given e.g. l. , 803, 874.
It is impossible to list them all.
Response 32: Thank you for pointing this out. We have made modifications to the format of the references.

Reviewer 3 Report
Comments and Suggestions for Authors
The paper covers an important and topical issue on increasing the drought resistance of maize. In my opinion the paper needs minor revision to clarify some aspects of the presented experiment.
These improvements are the following:
-the abstract is in my opinion well structured and comprehensive;
- the introduction of the paper successively brings to the attention the main research on maize, including other researches related to the improvement of maize resistance to drought; the aim of the paper is specified, also the hypothesis of the research; I think that at the end of the introduction it could be more clearly stated what is the novelty of the research in this paper in relation to the above researches (if there is any novelty?);
- table 1 - include in footnote a, b c -Different lowercase letters......
-embellishes the discussion with references, so it looks like a course lecture;
- all research methods must have references.
Author Response
Comments 1: the abstract is in my opinion well structured and comprehensive;
Response 1: Thank you for pointing this out.
Comments 2: the introduction of the paper successively brings to the attention the main research on maize, including other researches related to the improvement of maize resistance to drought; the aim of the paper is specified, also the hypothesis of the research; I think that at the end of the introduction it could be more clearly stated what is the novelty of the research in this paper in relation to the above researches (if there is any novelty?);
Response 2: Thank you for pointing this out. We agree with this comment. We have added relevant content in the introduction section. (Line94-Line111)
Comments 3: table 1 - include in footnote a, b c -Different lowercase letters......
Response 3: Thank you for pointing this out. We agree with this comment. We have made modifications according to your suggestions. (Line127, Line147)
Comments 4: embellishes the discussion with references, so it looks like a course lecture;
Response 4: Thank you for pointing this out. We agree with this comment. We added some references in the discussion. (Line427- Line428, Line431- Line436, Line449- Line452, Line466- Line467, Line477- Line479, Line500- Line503)
Comments 5: all research methods must have references.
Response 5: Thank you for pointing this out. We agree with this comment. We have added relevant references in the methodology section. (Line687, Line698, Line702, Line703, Line704)

Round 2
Reviewer 1 Report
Comments and Suggestions for Authors
Article have been improved substantially according to comments and suggestions provided. However, still needs further concern for betterment of the article.

Author Response
Thank you very much for taking the time to review this manuscript. We appreciate these valuable comments and suggestions very much. We have made detailed corrections in the manuscript corresponding to your suggestions and advice. Thank you for your time and consideration. According to the comments, we have studied the comments carefully and have made a correction which we hope meet with approval. At the same time, we have also made other changes.
Reviewer 2 Report
Comments and Suggestions for Authors
3323326

Author Response
Comments 1: Please note the letter designations in the figures, they should be in the top right corner;
Response 1: Thank you for pointing this out. We agree with this comment. We have made modifications according to your suggestions. To make it more visible and aesthetically pleasing, the letter designations was placed in the top left corner.
Comments 2: Figure 3. Please insert the letter designations of the following diagrams;
Response 2: Thank you for pointing this out. We agree with this comment. We have inserted letter designations in figure 3 (Line202)
Comments 3: There are still errors in subsection references, item 3, 4 etc. Please make a correction
Response 3: Thank you for pointing this out. We agree with this comment. We carefully checked and revised the references according to your suggestions.
Comments 4: General remark
The authors are responsible for the scientific and factual presentation of the research
presented, so please check the text in detail for content and editing;
Response 4: Thank you for pointing this out. We have made detailed modifications to the text for content and editing according to your suggestions.
